# Influenza Vaccination of Pregnant Women and Serious Adverse Events in the Offspring

**DOI:** 10.3390/ijerph16224347

**Published:** 2019-11-07

**Authors:** Alberto Donzelli

**Affiliations:** Executive Board of the Fondazione “*Allineare Sanità e Salute*”, 20122 Milan, Italy; adonzelli1@libero.it

**Keywords:** influenza vaccination, pregnant women, healthy-vaccinee bias, real-world vs. randomized evidence, offspring deaths, serious adverse events increase, informed consent

## Abstract

Pregnant women are increasingly considered a priority group for influenza vaccination, but the evidence in favor relies mainly on observational studies, subject to the “healthy-vaccinee bias”. Propensity score methods—sometimes applied—reduce but cannot eliminate residual confounding. Meta-analyses of observational studies show relative risks far from the thresholds that would confirm the efficacy of universal vaccination for pregnant women without needing randomized controlled trials (RCTs). Critical articles have shown that in the four RCTs investigating the outcomes of this vaccination, there was a tendency towards higher offspring mortality. In the largest RCT, there was a significant excess of presumed/serious neonatal infections, and also significantly more serious adverse events. Many widely acknowledged observational results (about hormone replacing therapy, vitamin D, omega-3 fatty acids, etc.) were confuted by RCTs. Therefore the international drive to consider this vaccination a “standard of care” is not justified yet. Moreover, there is the risk of precluding further independent RCTs for “ethical considerations”, so as “to not deny the benefits of influenza vaccinations to pregnant women of a control group”. Instead, before promoting national campaigns for universal vaccination in pregnancy, further large, independent, and reassuring RCTs are needed, even braving challenging a current paradigm. Until then, influenza vaccination should be offered to pregnant women only once open information is available about the safety uncertainties, to allow truly informed choices, and promoting also other protective behaviors.

## 1. Introduction

The WHO considers pregnant women the priority group for influenza vaccination [1], and there is global pressure to make it universal. However, a Cochrane review about influenza vaccination in healthy adults [2] found that the “NNV (number [of pregnant women] needed to be vaccinated to prevent one of them experiencing influenza) was 55 in mothers and 56 in infants”—resizing its effectiveness, the review continued: “The protective effect of vaccination in mothers and the newborn was very modest… smaller than the effect seen in other populations considered in this review…. We are uncertain of the protection provided to pregnant women against ILI [influenza-like illness] and influenza by the inactivated influenza vaccine, or this was at least very limited” [2].

The first condition for a universal vaccination campaign should be unambiguous effectiveness of the procedure. In the light of the Cochrane evidence-based conclusions, the WHO claim that pregnant women should be “the priority group for influenza vaccination” (first generated in 2012 [1], before the publication of the three largest RCTs on this topic) comes from an authoritative source, but does not seem adequately backed by evidence.

The effectiveness of the vaccine varies depending on how well it matches the circulating strains. However, the NNV in pregnant women in the Cochrane review does not differ much from the figure calculated in healthy adults, which summarizes all the available evidence, including 52 clinical trials of more than 80,000 people, conducted over single influenza seasons, in the Americas and in Europe, in a timeframe of over 40 years. Regardless, it is impossible even to forecast the efficacy in a specific year, hence, the universal vaccination of pregnant women will always have the same universal denominator, regardless of the match, that can only be verified afterwards.

The second condition for universal vaccination should be proven safety. The evidence of the effectiveness and safety of influenza vaccination in pregnant women still relies mainly on observational studies.

## 2. Objective, Materials, and Methods

The objective of this perspective article is to present a novel viewpoint on the topic of influenza vaccination for pregnant women. It presents original data as well as personal opinions, analyzing also new observational evidence, seeking plausible alternative explanations for some favorable outcomes in the offspring of vaccinated pregnant women, and quantifying the limited benefits in their offspring in recent meta-analyzes of observational studies (pooling two trials at best). It adds some new evidence from the RCTs, to date, raising new questions about the safety of influenza vaccination during pregnancy. The main articles analyzed are summarized in Table 1.

Finally, this perspective proposes some coherent implications of health policies.

## 3. Results and Their Discussion

### 3.1. Observational Studies and Healthy-Vaccinee Bias

The observational design is subject, among others, to the “healthy adherer bias”: subjects who adhere to preventive therapies are at the same time more likely to engage in healthy lifestyles than patients not adhering to such strategies [3,4]. A healthy lifestyle includes diet, exercise, less alcohol intake and risky behaviors, and the search for better health care. These features—difficult to capture in administrative databases—are associated with morbidity and mortality outcomes in observational studies.

Similarly, in the vaccine field, there is the “healthy-vaccinee bias”, that leads to the overestimation of the vaccine’s effectiveness and safety [5,6]. Influenza vaccination of the elderly provided a clear example of this bias [7].

Systematic investigation of this bias and of its opposite “confounding-by-indication bias” [8] showed that statistical adjustment can fairly well correct the confounding-by-indication bias, but cannot adequately compensate the healthy-vaccinee bias. This bias may be strong in pregnant women: more self-disciplined and educated women generally have healthier behaviors and are more adherent to recommendations by physicians, obstetricians, and health authorities [9], making them --much more likely to receive vaccination [9]; unlike socio-economically and culturally less advantaged women [10,11]. Other biases of the observational studies were highlighted in 2016 [12], and their correction has removed the significance of further alleged benefits.

Previous publications have illustrated this bias, showing structural differences between the cohorts of pregnant women adhering or not to influenza vaccination [5,6]. A CDC study [13] found unvaccinated women somewhat less affected by “high risk diseases”, which translated into slightly less likelihood of being vaccinated: 46.3% versus 54.0% (7.7% absolute points difference); however, this characteristic was overwhelmed by the heavy disadvantage of the unvaccinated from educational and socio-economic angles [13].

This does not constitute a special case. Some European studies have confirmed important structural differences between the cohorts of vaccinated and unvaccinated pregnant women. A French prospective cohort study [14] found that the cohort of unvaccinated mothers presented multiple major disadvantages. The determinants associated with non-vaccination in a multivariate logistic regression included geographic origin: Sub-Saharan African origin gave an adjusted Odds Ratio aOR of 5.4 (95% CI 2.3–12.7), North African origin had aOR 2.5 (1.3–4.7), and Asian origin aOR 2.1 (1.7–2.6), compared to women of French and European origin. Then too, compared to managers and intellectual professionals, the categories of farmers, craftsmen and tradesmen had an aOR 2.3 (2.0–2.6), intermediate professionals aOR 1.3 (1.0–1.6), employees and manual workers aOR 2.5 (1.4–4.4). The probability of not receiving pandemic flu vaccine was lower among women who stopped smoking before or early during pregnancy, with aOR 0.6 (0.4–0.8) compared with no smokers, while current smokers had an aOR 1.2 (0.8–1.8).

Quite likely, the worse socio-economic and behavioral conditions of unvaccinated women could explain the worse outcomes of their children, without having to appeal to the explanation of missed vaccination. Having at least one associated co-morbidity acted in the opposite direction, but to a lesser extent and without reaching statistical significance: aOR 1.2 (0.9–1.5) [14]. Even a significant obstetric history tended to raise the pregnant women’s likelihood of influenza vaccination, but the difference did not reach statistical significance.

Some of these characteristics are not measured in pharmacological databases, or are difficult to capture even when applying a propensity score weighting, because it is impossible to include unmeasured or unknown confounders in the propensity scores [15]. The adherence to vaccination can also be linked to more trust in the proposed intervention, and this in turn might lead to better outcomes to some extent [6]. Moreover, the propensity scores and their reliability depend closely on how they are built, hence, their authors should always provide independent reviewers with details of the methods employed, so that they can replicate the process, and one can never dismiss the possibility of residual confounding of findings.

A systematic review of safety outcomes associated with influenza vaccination during pregnancy [16] did not include randomized controlled trials (RCTs), though one of them was cited. Later systematic reviews were done and we report the results of the two latest.

A systematic review of safety of inactivated influenza vaccines in pregnancy for birth outcomes [17] included 39 observational studies (25 retrospective and nine prospective cohort studies, three case-control and two cross-sectional studies) and one RCT [18]. The adjusted Odds Ratios (aOR) were: for preterm birth (PTB) 0.87 (0.78–0.96), for low birth weight (LBW) 0.82 (0.76–0.89), congenital abnormality 1.03 (0.99–1.07), small for gestational age (SGA) 0.99 (0.94–1.04), and stillbirth 0.84 (0.65–1.08).

The last systematic review [19] did a Bayesan meta-analysis, reviewing the largest number of articles of reasonable quality. It found no significant decrease for any of the following adverse birth outcomes: PTB (odds ratios [OR] 0.945, 95% credible intervals [CrI]: 0.736–1.345, P = 73.3%), low birth weight (OR 0.928, 95% CrI: 0.432–2.112, P = 76.7%), small for gestational age (OR 0.971, 95% CrI: 0.249–4.217, P = 63.3%), congenital malformation (OR 1.026, 95% CrI: 0.687–1.600, P = 38.0%), and fetal death (OR 0.942, 95% CrI: 0.560–1.954, P = 61.6%). The conclusion is: “results showed evidence of null associations between maternal influenza vaccination and adverse birth outcomes”. After adjusting for season at the time of vaccination and the country’s income level, the summary estimates including only *cohort* studies showed a significant decrease limited to fetal deaths; however, this is an implicit admission that in fact fetal deaths showed a tendency to be *higher* in the two RCTs considered [18,20], as was indeed the case.

The Lancet has just reported an interesting debate on why real-world searches cannot substitute for RCTs in establishing efficacy [21]. The RCT may not be needed to assess causality in the rare situations in which “Confounding is unlikely to underlie relationships with extreme relative risks such as those less than 0.25 or greater than 4 [21].” Instead, “relationships with relative risks varying between 0.5 and 2 are the ones most commonly reported in analyses of real-world data, and are those most susceptible to unaccounted-for confounding” [21]. In these situations RCTs are “irreplaceable”. The conclusion is “In the absence of randomization, analyses of most observational data from the real world, regardless of their sophistication, can only be viewed as hypothesis-generating [21].” A letter from three Cochrane reviewers reported even more stringent criteria: “Other studies have suggested different thresholds of an RR of 10 or higher, 2 or 5 or higher (or RR < 0.2) to avoid an RCT [22].” Each of these thresholds would certainly call for additional RCTs in the face of results of systematic reviews of observational studies such as the two reported above ([17,19] when excluding the two RCTs).

A recently published retrospective cohort study [23] found no association between 2009 women given pandemic H1N1 vaccines in pregnancy and most five-year pediatric health outcomes. The study also cited the four RCTs commented below, but failed to mention some articles [5,6,24] which had questioned the reassuring conclusions of their authors, not supported by their own data. This omission occurred though one of the main authors [23] recently published a Comment [25] to these critics, receiving a Reply letter [24]. An Editorial of the retrospective study [23] concludes: “Vaccination of pregnant women saves lives” [26], but the RCTs tell a different story (Table 1).

### 3.2. RCTs: Some New Evidence

Meta-analyses of observational studies (such as [17]) show relative risks far from the thresholds that would support the efficacy of this vaccination in the absence of RCTs [21,22]. However, four RCTs are currently available. After the first small RCT in Bangladesh [27], the Bill & Melinda Gates Foundation funded three large blinded RCTs [30], partly to overcome the validity issues of observational evidence. One, the Matflu [18], was in an upper-middle-income country. It is considered “at low risk of bias” by Cochrane reviewers [2], with a number to vaccinate (NNV) 55 to avoid one influenza in mothers. Two other RCTs were in low-income countries [20,28], where the expected benefits were greater.

Besides the placebo-controlled Matflu RCT [18], included in the Cochrane review [2], one can also take into account the placebo-controlled Nepalese RCT [20], in which the NNV (not specified) seems about 20; and the other two active-controlled RCTs: the small RCT in Bangladesh [27], NNV 17 (control group with 23-valent pneumococcal vaccine), and the much larger RCT in Mali [28], NNV 99 (control group quadrivalent meningococcal vaccine). The overall NNV is not far from the Cochrane estimate, close to the NNV for healthy adults [2].

In the placebo-controlled RCTs, the best evidence showed “very modest” efficacy of vaccination at the population level [2], and an excess of maternal local adverse effects [5,24]. Moreover, in the influenza-vaccinated women the offspring mortality tended to be higher than in the control groups. The overall serious adverse events (SAEs) tended to be more numerous, as summarized in an infographic [24]. In the larger RCT the excess of SAEs in the offspring was statistically significant, according to my personal calculation: the number of livebirths per vaccine group was 2064 in vaccinated women, 2041 in the active control group; the total SAEs were 225 (10.90%) and 175 (8.57%) respectively in Table S7 [28]. Therefore, the RR is 1.27; CI 95% 1.05–1.53; NNH (number needed to harm) 42.98.

A sensitivity analysis can exclude from the SAE count the major congenital malformations, six (influenza group) and four (control group), not correlated with vaccinations received by the 28th week of gestation. The analysis again shows a significant excess of SAEs in the vaccinated group: RR 1.27; CI 95% 1.05–1.53; NNH 44.80.

The abstract reports: “Presumed neonatal infection was more common in infants in the trivalent inactivated influenza vaccine group than in those in the quadrivalent meningococcal vaccine group (n = 60 vs. n = 37; *p* = 0.02)” [28], without clearly specifying that:➢These presumed/neonatal infections were SAEs, not *generic* infections➢the total SAEs (which included infant deaths: 52 vs. 37 in the control group) were also significantly more in the offspring of vaccinated mothers➢the total SAEs (not just the individual categories of SAEs) are indeed a “hard” standard outcome.

An emerging international debate around the possible adverse fetal outcomes of this vaccination in pregnant women has led to a reanalysis of the Matflu trial [29]. The authors’ conclusion was: “We did not find a beneficial effect of trivalent inactivated influenza vaccine during pregnancy on adverse fetal outcomes.” Indeed, the vaccine’s efficacy tended to be worse for every outcome: fetal death (efficacy −21.2% [−150.8, 41.4]), LBW (−11.1% [−42.3, 12.5]), SGA (−9.9% [−35.6, 11.0]), or PTB (−21.3% [−60.5, 8.3]). In analysis restricted to infants of mothers exposed to an influenza season, the tendency was equally unfavorable [29].

The mechanism of hypothetical harms is unclear. Possibly there is an inflammatory stress linked to this vaccination [31,32,33]: The inflammatory stress of an influenza is certainly greater, but the tradeoff is between 1 influenza and 55 vaccinations. An intriguing study [34] covered 1,791,520 Swedish children born in a wide time-span, and found that fetal exposure to a maternal infection was associated with a greater long-term risk of neuropsychiatric disease, even for mild maternal urinary tract infections (UTIs). In the discussion, the authors state: “we found compelling evidence that fetal exposure to infection (or inflammation) when the mother was hospitalized increased the risk for the child… during childhood or adulthood… irrespective of whether the exposure was a maternal severe infection… or UTI during pregnancy... results were robust to adjustment for a moderate unknown confounder” [34]. In the absence of strong evidence of benefit and long-term safety, the precautionary principle should suggest avoiding any deliberate inflammatory stimuli in the vulnerable stage of pregnancy, even if the authors [34]—within the current paradigm—conclude by proposing a primary prevention based on vaccinations, or anti-inflammatory therapies.

It seems there is a clear need for further large, pragmatic RCTs by independent institutions and researchers, with a long extension [35]. A previous article suggested a way to overcome the “ethical concerns” fairly, by recruiting only women who were still hesitant even after having received balanced information about the pros and cons of vaccination during pregnancy, at the present state of knowledge [35].

### 3.3. General Comments

Since observational studies in pregnant women (usually quite young and healthy) are prone to the healthy-vaccinee bias, to promote a preventive pharmacological intervention, particularly in the vulnerable period of a pregnancy, public health services should not rely only or primarily on observational evidence [5,6,24]. Even more so if the (insufficient) safety evidence from existing RCTs shows a tendency to move in an opposite and alarming direction.

With the current state of knowledge, the international drive and proclaimed urgency to consider influenza vaccination of pregnant women a “standard of care” is unsupported, and precludes new independent RCTs that could clarify the issue, with the argument of “ethical considerations in order not to deny the benefits of vaccination to the placebo group”. This widespread attitude refers to what was described by epistemologists such as Kuhn [36], when anomalies emerge that could falsify a dominant paradigm. At this stage, the majority of the scientific community can react with intolerance [36], even choosing to ignore the facts that could open up a paradigm crisis, thus precluding the possibility of reaching more comprehensive syntheses. Instead, I agree with Popper’s conclusion [37] that the scientific attitude is the critical attitude, which does not search (only) for verifications, but for crucial controls potentially able (even) to falsify a current theory, when anomalies could put it into question.

We should beware of the results of observational studies not confirmed by corresponding RCTs. An example is hormone replacement therapy (HRT) to prevent chronic conditions in postmenopausal women. The estimated calculation of total harms vs. benefits for 10,000 person-years associated with HRT are: Estrogen + progestin 971 harms vs. 65 benefits; estrogen only 1.329 harms vs. 82 benefits [38,39]. Other examples are vitamin D supplements “for health” [40], or omega-3 fatty acid supplements for cardiovascular diseases [41].

## 4. Conclusions

The RCTs available today raise safety signals of possible life-threatening events in the offspring of influenza-vaccinated pregnant women. Before promoting universal vaccination in pregnancy, further large, independent, and reassuring RCTs are needed, refraining from unsubstantiated “ethical reasons” for not allowing them. The safety variables to assess are all the solicited adverse events, with special attention to SAEs, in a wide time window, plus many years of follow-up after the double-blind interruption, in order to detect even subtle differences in any direction emerging in the long term.

Meanwhile, vaccination could be offered to pregnant women, informing them in a balanced way (including the fact that the majority of ILI are not caused by influenza viruses and are not preventable by vaccines), to allow real informed choice and consent. The statement that the offspring of unvaccinated mothers could suffer severe consequences should be avoided or balanced, since current RCTs show that even the opposite might be true. Moreover, other protective behaviors, previously detailed [5], should also be promoted.

## Figures and Tables

**Table 1 ijerph-16-04347-t001:** Main articles reviewed and discussed in this perspective article.

Author (Surname)	Name, Article Year [Ref. Number] Participants	Journal	Important Findings
Kahn	Influenza Tdap vaccination coverage among pregnant women, 2018 [13]	*MMWR*	This U.S. survey shows a quantitative view of the size of the possible *healthy-vaccinee bias.* Its inadequate correction might explain the worse outcomes of the offspring of unvaccinated women
Freund	COFLUPREG, 2011 [14]	*PLoS ONE*	This French survey shows a quantitative view of the size of the *healthy-vaccinee bias* in a European country. Inadequate correction of this bias might explain the differences in the offspring’s outcomes
Giles	The safety of IIV in pregnancy for birth outcomes: a systematic review, 2018 (39 observational studies and one RCT) [17]	*Hum Vaccin Immunother*	Only preterm birth and low birth weight showed a significant advantage in offspring of vaccinated mothers (aOR 0.87 and 0.82, respectively). No other outcome showed any significant differences
Jeong	Effects of maternal influenza vaccination on adverse birth outcomes: a systematic reviews and Bayesian meta-analysis, 2019 (41 cohort, 5 case-control studies, and two RCTs) [19]	*PLoS ONE*	This most recent and comprehensive systematic review found no significant decrease in any of the adverse fetal outcomes: preterm birth, low birth weight, small for gestational age, congenital malformation, fetal death
Madhi	Matflu, 2014 [18]	*New Engl J Med*	NNV = 55 (mothers), 56 (children)
			**Severe safety signals** (no significant differences):
			(influenza vaccinated vs. placebo)
			Maternal deaths: 2 vs. 0
			Maternal hospitalization for any infection: 16 vs. 7
			Miscarriages: 3 vs. 5 (4?)
			Stillbirths: 15 vs. 9
			Infant deaths: 15 vs. 21
			Infant hospitalization for sepsis in the first 28 days of birth: 16 vs. 10
			Infant hospitalization for meningitis in the first 28 days: 6 vs. 2
Steinhoff	Nepalese trial, 2017 [20]	*Lancet Infect Dis*	NNV = about 20 (not specified)
			**Severe safety signals** (no significant differences):
			(influenza vaccinated vs. placebo)
			Maternal deaths: 3 vs. 5 in the abstract (2 vs. 5 in *STable*2)
			Miscarriages: 5 vs. 3
			Stillbirths: 33 vs. 31
			Infant deaths: 61 vs. 50 in the abstract (60 vs. 51 in *STable*2)
			Congenital defects: 20 vs. 18
Zaman	Mother’sGift project, 2008 [27]	*New Engl J Med*	NNV = 17
			**Severe safety signals** (no significant differences):
			(influenza vaccinated vs. pneumococcal)
			Maternal deaths: 0 vs. 0
			Fetal + infant deaths: 4 vs. 2
Tapia	Mali trial, 2016 [28]	*Lancet Infect Dis*	NNV = 99
			**Severe safety signals** (no significant differences, except presumed serious neonatal infections and total SAEs):
			(influenza vaccinated vs. meningococcal)
			Maternal deaths: 1 vs. 0 (in Table S6: SAEs in participating women ‘at any time after vaccination until 6 months post-partum’), but 2 vs. 3 (in the article)
			Stillbirths: 24 vs. 30
			Infant deaths: 52 vs. 37
			Presumed serious neonatal infections: 60 vs. 37 (*p* = 0.02)
			Total SAEs: 225 vs. 175 (*p* = 0.01); excluding major congenital malformations 219 vs. 171 (*p* = 0.015)
Simões	Matflu (reanalysis) 2019 [29]	*Vaccine*	**Serious safety signals** (no significant differences):
			(influenza vaccinated vs. placebo)
			Fetal death: 16 vs. 13 (RR 1.21)
			Preterm birth: 100 vs. 80 (RR 1.22)
			Low birth weight: 123 vs. 107 (RR 1.21)
			Small for gestational age: 156 vs. 138 (RR 1.21)

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
