# Peer review of "Influenza Vaccination of Pregnant Women and Serious Adverse Events in the Offspring"

_ijerph, 2019, doi:10.3390/ijerph16224347_

Round 1
Reviewer 1 Report
In this manuscript the author discusses the need for better evidence, via properly designed randomized controlled trials, about the safety of influenza vaccination during pregnancy. I have several issues with this manuscript: Line numbers have been removed from the text making review difficult. The article straddles the line between a review and an editorial. References seem "cherry-picked" to support the author's viewpoint with few competing Cochrane analyses and/or independent studies presented. Language used throughout the manuscript is not that typically found in scientific articles. For example, the sentence "the evidence favoring the effectiveness and the safety of influenza vaccination in pregnant women relies essentially on observational studies". While it may be fair to say that there are far greater number of observational studies, the use of essentially comes across as an editorial comment. Discussing one's own questions about vaccine safety and proposing implications of health policies do not qualify as Methods per se. The author quotes his own previous publications to make the claim that "the observational design is subject to the "healthy-vaccinee bias", that overestimates the efficacy and safety of the vaccine [3,4]." This is unacceptable. Other independent references supporting this claim about the relationship between observational studies and flu vaccine efficacy and safety are needed. The author's discussion of the Matflu trial is flawed. First, the wrong citation is used to discuss the reanalysis (ref 23, not 22). Second, the purpose of the reanalysis was not so much to evaluate vaccine safety as it was to determine whether IAV vaccination provided benefits related to the prevention of fetal death, low birth weight, small for gestational age birth and preterm birth. Indeed, the paper did not link these outcomes to any conditions or infections. If anything, they were observations! Third, while the author includes data to support his belief that IAV vaccination poses a risk to newborns he fails to include the P values calculated from the Matflu study. Differences in fetal death, low birth weight, small for gestational age birth and preterm birth were not statistically significant. The author's discussion of Christian et al, Am J Repro Immunol, 2013 is also flawed, misconstrued. The purpose of that paper was to investigate whether inflammatory responses to vaccination would be attenuated in pregnant women. The authors of that paper found that inflammation following vaccination was transient and mild. These authors also remarked "... these results are consistent with prior data showing that inflammatory responses to TIV vaccination are mild and transient in pregnant women, supporting the safety of vaccination." One could even argue that the inflammatory reponse following vaccination is far less than that observed during a live flu infection. While I respect the author's efforts in analyzing the literature for observational and RCT studies, this article is not significantly different, nor novel, compared to his prior publications (ref #2 and #3).Author Response
R1
Comments and Suggestions for Authors
In this manuscript the author discusses the need for better evidence, via properly designed randomized controlled trials, about the safety of influenza vaccination during pregnancy. I have several issues with this manuscript:
Line numbers have been removed from the text making review difficult.
--- The line numbers have been inserted.
The article straddles the line between a review and an editorial.
--- I agree: this is what I consider a perspective article, presenting a new viewpoint on influenza vaccination for pregnant women. The perspective typically presents original data as well as personal opinions, usually about 3000 words (the first version of the article was about 2700 words). I apologize for not having clarified this in the original cover letter, and hope it is now clear.
References seem "cherry-picked" to support the author's viewpoint with few competing Cochrane analyses and/or independent studies presented.
--- As I presented my viewpoint, it seemed logical to provide references to support it: a perspective article is of course not a systematic review (though without overlooking any valid data - to my knowledge – that could falsify my view, but disregarding differing opinions and conclusions, when not based on consistent data).
In the section about observational studies, I cited [reference 5], the most recent CDC report that gives a quantitative idea of the healthy-vaccinee bias; and I cite the only European study (to my knowledge) with similar characteristics [ref. 6].
As regards systematic reviews of observational studies of offspring outcomes related to influenza vaccination of the mother in pregnancy, I explicitly cite and analyze the two most recent comprehensive ones. The first [ref. 10] found a significant but quite modest benefit only for PTB (aOR 0.87) and LBW (aOR 0.82), without significant benefits for SGA, stillbirth and congenital abnormalities. The latest systematic review [ref. 11] did a Bayesan meta-analysis, reviewing as many articles of reasonable quality as possible. In the offspring of vaccinated women it found no significant reductions in the following adverse birth outcomes: PTB, LBW, SGA, congenital malformation, and fetal death. The conclusion was: “results showed evidence of null associations between maternal influenza vaccination and adverse birth outcomes”.
---In the section on RCTs, I analyze the four RCTs I found.
As suggested by Reviewer 3, I have now summarized this information in Table 1.
Language used throughout the manuscript is not that typically found in scientific articles. For example, the sentence "the evidence favoring the effectiveness and the safety of influenza vaccination in pregnant women relies essentially on observational studies". While it may be fair to say that there are far greater number of observational studies, the use of essentially comes across as an editorial comment.
--- A Perspective does indeed share some similarities with Commentaries, but it is not invited like an editorial comment. It includes not only personal viewpoints but also some new data, and can be longer, because Commentaries are usually around 1000-1500 words. However, I have deleted ‘essentially’ and used only ‘mainly’ because the evidence from all the four RCTs does not give reassurance about the safety of this vaccination, and shows tendencies in a different direction.
Discussing one's own questions about vaccine safety and proposing implications of health policies do not qualify as Methods per se.
--- A Perspective article has of course fewer constraints on how the methods are described than a research article or a systematic review. However, I have now described the methods more explicitly.
The author quotes his own previous publications to make the claim that "the observational design is subject to the "healthy-vaccinee bias", that overestimates the efficacy and safety of the vaccine [3,4]." This is unacceptable. Other independent references supporting this claim about the relationship between observational studies and flu vaccine efficacy and safety are needed.
--- In the two articles cited I have amply discussed the healthy-vaccinee bias, with references, and I did not want to repeat them. However, I have now documented the statement better.
The author's discussion of the Matflu trial is flawed. First, the wrong citation is used to discuss the reanalysis (ref 23, not 22).
--- Maybe there is a misunderstanding here: the ref. 22 I sent was “Simões EAF, …. Madhi SA. Vaccine 2019, 37, 5397–5403”, while 23 is “Christian… Vaccine 2011, 29, 8982–8987”. However, with the new references in response to your previous request, I had to change the reference numbers and hope they are now clear.
Second, the purpose of the reanalysis was not so much to evaluate vaccine safety as it was to determine whether IAV vaccination provided benefits related to the prevention of fetal death, low birth weight, small for gestational age birth and preterm birth. Indeed, the paper did not link these outcomes to any conditions or infections. If anything, they were observations!
--- I have changed the word “safety” to read “possible adverse fetal outcomes”, the terms used by Simões EAF,…. Madhi SA. (Vaccine 2019) in their title. Re-analysis of this high-quality trial indicated that IAV vaccination did not provide any benefit related to the prevention of fetal death, LBW, SGA and PTB, but in fact a tendency to harm for all four outcomes. Therefore, it seems to bolster up my theory: if the purpose was to obtain substantial benefits for the offspring, these are not proven, so why bear the costs of universal vaccination in the face of other cost opportunities/public health priorities? If the purpose was the mothers’ health, this too is not proven.
In Madhi’s trial there were two deaths among the vaccinated mothers, none with the placebo (and there were more maternal hospitalizations for any infection among the vaccinated women: 16/1062, vs 7/1054 controls). In Steinhoff’s trial there were 3 deaths (stated in the abstract) versus 5 for placebo; in Tapia’s trial 1 death versus 0 in Table S6 (or 2 versus 3 in the article). No mothers died in Zaman’s trial. Therefore, there was no definitive advantage for maternal health.
Third, while the author includes data to support his belief that IAV vaccination poses a risk to newborns he fails to include the P values calculated from the Matflu study. Differences in fetal death, low birth weight, small for gestational age birth and preterm birth were not statistically significant.
--- I originally wrote: “in tendency the vaccine efficacy was worse for every outcome: …”. This has now been revised to read ‘there was a tendency for the vaccine to be less efficacious for every outcome’. The inclusion of the P values seemed burdensome, but if the Reviewer wishes, I would have no problem giving them.
The author's discussion of Christian et al, Am J Repro Immunol, 2013 is also flawed, misconstrued. The purpose of that paper was to investigate whether inflammatory responses to vaccination would be attenuated in pregnant women. The authors of that paper found that inflammation following vaccination was transient and mild. These authors also remarked "... these results are consistent with prior data showing that inflammatory responses to TIV vaccination are mild and transient in pregnant women, supporting the safety of vaccination." One could even argue that the inflammatory reponse following vaccination is far less than that observed during a live flu infection.
--- I faithfully reported Christian's data, but I did not feel obliged to share her conclusion. The data led me to observe that “the inflammatory stress of an influenza vaccination is certainly greater, but the trade-off is between 1 influenza and 55 vaccinations”. I also cited a recent article (al-Haddad, 2019) where the authors state: “we found compelling evidence that fetal exposure to infection (or inflammation: the authors use the term “inflammation” in the Discussion)… irrespective of whether the exposure was a maternal severe infection… or UTI during pregnancy. ... results were robust to adjustment for a moderate unknown confounder”.
While I respect the author's efforts in analyzing the literature for observational and RCT studies, this article is not significantly different, nor novel, compared to his prior publications (ref #2 and #3).
--- My earlier cover letter highlighted the seven points of novelty compared to my previous publications: I report them below:
“ This manuscript contains many new analyses, new data, and new concepts about the debated issue of the safety of influenza vaccination in pregnant women, raised thanks to our contribution. Neither the manuscript nor any parts of its content are currently under consideration in another journal.
It contains:
the analysis of a French prospective cohort study (Freund, 2011), confirming for the first time in an European population the same structural differences between vaccinated and unvaccinated pregnant women already seen and analyzed in the CDC study in the US (Kahn, 2018), showing a strong healthy-vaccinee bias; the discussion of the systematic reviews and meta-analyses of observational studies about influenza vaccination of pregnant women and outcomes in their offspring, particularly the two most recent and comprehensive reviews. The one published in 2018 showed only two significant aORs: for preterm birth 0.87 and for low birth weight 0.82, without other significant benefits. The Bayesan meta-analysis published in August 2019 showed no significant decrease for any adverse fetal outcome; the current debate in The Lancet, stating that in the absence of RCTs analyses of real-world data can assess causality only with extreme RRs (less than 0.25 or greater than 4, or even greater than 10). These thresholds are far from those found in the above meta-analyses of observational studies; the overlooked finding of a significant excess of total SAEs in the largest of the RCTs about influenza vaccination in pregnant women (Tapia), with my calculation of the exact RR.
This was a new finding also for me, because an epidemiologist, reader of a previous article, wrote to me that the correction for multiple comparisons could perhaps take off significance even from the excess of serious “presumed/neonatal infections” in the Tapia’s RCT in Mali. This prompted me to check carefully Table 7 (in the Supplementary webappendix), noticing that the total SAEs (not subject to correction for multiple comparisons) did indeed show a significant excess. This fact, not highlighted by the authors, not even in the webappendix, should instead be brought to the attention of the scientific community.
new data from the authors of the Matflu trial, confirming in the offspring of influenza vaccinated mothers the tendency to harm for every birth outcome analysed; three famous examples of observational “evidence” dis-confirmed (rightly) by corresponding RCTs (HRT, vitamin D, omega-3 fatty acids), with important methodological lessons to be carefully evaluated; a fundamental reason not to define the universal influenza vaccination of pregnant women a “standard of care”, without adequate evidence: indeed this statement could create a vicious circle, precluding other large and independent RCTs, which could clarify the important safety issues raised by the four available trials [I also have added some epistemological considerations]. ”
I do feel that the social responsibility of a medical journal and of a reviewer might well take into account what is happening all over the world. In all countries there is pressure to give priority to the universal vaccination of pregnant women, stressing only the possible pros, and not providing balanced information about the cons and the serious persisting uncertainties. In addition, the pressure from the scientific community to consider this vaccination a ‘standard of care’ precludes for ‘ethical considerations’ other large independent RCTs, which could clarify the safety issues raised by the trials to date. This can prevent the progress of scientific knowledge.
Reviewer 2 Report
Alberto Donzelli shows a paper in which he has reviewed some of the scientific papers that have analyzed the vaccine effectiveness of the flu vaccine in pregnant women and the effects on offspring. Although a review of these articles is interesting, the author draws a series of conclusions that are not supported at any time by the articles he mentions, offering a very contradictory view of the recommendations currently made by WHO for the use of influenza vaccine in risk groups.
The article in general is poorly written, difficult to understand and uses the information of the cited manuscripts in a biased manner. There are editing errors, both in word size and format throughout the manuscript, as well as errors in bibliographic references everywhere, which prevent following the reading of the text in a clear way.
The part of material and methods is non-existent, and it does not allow to know what methodology of analysis of the results of other articles the author has followed. The format of this manuscript seems more suitable for a "letter to the editor" or for an "opinion piece", but not for a larger article as intended in this case
.
The article needs an extensive edition in English.
Author Response
R2
Alberto Donzelli shows a paper in which he has reviewed some of the scientific papers that have analyzed the vaccine effectiveness of the flu vaccine in pregnant women and the effects on offspring. Although a review of these articles is interesting, the author draws a series of conclusions that are not supported at any time by the articles he mentions,
--- The reviewer is not specific, so unfortunately I fail to see what ‘lack of support’ he refers to. I suppose he is noting the absence of statistically significant harms in the vaccinated groups in any of the four trials. This is true, but it is also true that – applying the appropriate [statistical] corrections – even the benefits of this vaccination reported in the reviews of observational studies lose significance: it happened in the most comprehensive recent systematic review (Jeong, PlosONE 2019).
Therefore, the four main questions are:
1) why impose the economic burden of universal vaccination if its benefits are not firmly established?
2) why consider this an immunization priority, compared to other public health interventions of much greater effectiveness and cost-effectiveness, unencumbered by so many uncertainties?
3) why disregard the safety signals? For safety issues one must also consider the information provided by tendencies, even those without statistical significance. Here this is important because of the consistent tendencies to serious damages in each of the four RCTs.
4) why not leave open at least a scientific debate, while dozens of continuously published articles push for vaccinating pregnant women, and health care professionals promote this vaccination as absolutely safe, stating that its avoidance is dangerous for the mothers and the offspring, despite current uncertainties?
offering a very contradictory view of the recommendations currently made by WHO for the use of influenza vaccine in risk groups.
--- It is true that the article questions the WHO’s current recommendations, but they were formulated in 2012, when the three large RCTs on this topic had not yet been published, and the corrections of biases in observational studies were totally inadequate. The credibility of the health organisations is not safeguarded by reiterating their recommendations, even when they need to be re-assessed in the light of new evidence. If Karl Popper were alive, I think he would want to leave this debate open.
The article in general is poorly written, difficult to understand
--- The article has been revised by an English author’s editor in Milan.
and uses the information of the cited manuscripts in a biased manner.
--- I am available and interested in discussing your critical remarks, but I would be grateful if you could kindly be more specific.
There are editing errors, both in word size and format throughout the manuscript,
--- The article has been revised by an English author’s editor in Milan. I hope the new version no longer suffers from errors in word size.
as well as errors in bibliographic references everywhere, which prevent following the reading of the text in a clear way.
--- I have checked and corrected the references.
The part of material and methods is non-existent, and it does not allow to know what methodology of analysis of the results of other articles the author has followed. The format of this manuscript seems more suitable for a "letter to the editor" or for an "opinion piece", but not for a larger article as intended in this case.
--- I apologize for not having clarified this properly in the original cover letter. It is not a research article or a systematic review, but a Perspective article, presenting a new viewpoint on the topic of influenza vaccination for pregnant women. A Perspective typically presents original data as well as personal opinions, with about 3000 words (the first version of the article was about 2700 words. The new version is a bit longer, to satisfy the Reviewers’ requests). A Perspective article has fewer constraints on the methods than a research article or a systematic review. I have now described the methods more explicitly.
.
The article needs an extensive edition in English.
--- The article has been revised by an English author’s editor in Milan.
Reviewer 3 Report
The paper addresses an extremely important topic - side effects of the flu shot for pregnant women. However, the analysis needs more structure: how was the literature search conducted? Which articles were included?
A table of the articles reviewed would be helpful in adding structure. The table should include author, name and year of article, journal, and important findings.
The English was very difficult to understand. I sense the paper has more to offer than what I was able to grasp, because I could not understand what the author was trying to say.
Author Response
R3
The paper addresses an extremely important topic - side effects of the flu shot for pregnant women. However, the analysis needs more structure: how was the literature search conducted? Which articles were included?
--- I apologize for not having clarified this properly in the original cover letter. It is not a research article or a systematic review, but a Perspective article, presenting a new viewpoint on the topic of influenza vaccination for pregnant women. A Perspective typically presents original data as well as personal opinions, with about 3000 words (the first version of the article was about 2700 words. The new version is a bit longer, to satisfy the Reviewers’ requests). A Perspective article has fewer constraints on the methods than a research article or a systematic review. I have now described the methods more explicitly.
A table of the articles reviewed would be helpful in adding structure. The table should include author, name and year of article, journal, and important findings.
--- Thank you for the suggestion. I have added a table (below).
The English was very difficult to understand. I sense the paper has more to offer than what I was able to grasp, because I could not understand what the author was trying to say.
--- The article has been revised by an English author’s editor in Milan.
Table 1 – Main articles reviewed and discussed in this perspective article
|
Author (surname) |
Name, article year [ref. number] participants |
Journal |
Important findings |
|
Kahn |
Influenza and Tdap vaccination coverage among pregnant women, 2018 [5] |
MMWR |
This U.S. survey shows a quantitative view of the size of the possible healthy-vaccinee bias. Its inadequate correction might explain the worse outcomes of the offspring of unvaccinated women |
|
Freund |
COFLUPREG, 2011 [6] |
PLoS ONE |
This French survey shows a quantitative view of the size of the healthy-vaccinee bias in a European country. Inadequate correction of this bias might explain the differences in the offspring’s outcomes |
|
Giles |
The safety of IIV in pregnancy for birth outcomes: a systematic review, 2018 (39 observational studies and one RCT) [9] |
Hum Vaccin Immunother |
Only preterm birth and low birth weight showed a significant advantage in offspring of vaccinated mothers (aOR 0.87 and 0.82, respectively). No other outcome showed any significant differences |
|
Jeong |
Effects of maternal influenza vaccination on adverse birth outcomes: a systematic reviews and Bayesian meta-analysis, 2019 (41 cohort, 5 case-control studies, and two RCTs) [11] |
PLoS ONE |
This most recent and comprehensive systematic review found no significant decrease in any of the adverse fetal outcomes: preterm birth, low birth weight, small for gestational age, congenital malformation, fetal death |
|
Madhi |
Matflu, 2014 [10] |
New Engl J Med |
· NNV = 55 (mothers), 56 (children) Severe safety signals (no significant differences): (influenza vaccinated vs. placebo) · Maternal deaths: 2 vs. 0 · Maternal hospitalization for any infection: 16 vs. 7 · Miscarriages: 3 vs. 5 (4?) · Stillbirths: 15 vs. 9 · Infant deaths: 15 vs. 21 · Infant hospitalization for sepsis in the first 28 days of birth: 16 vs. 10 · Infant hospitalization for meningitis in the first 28 days: 6 vs. 2 |
|
Steinhoff |
Nepalese trial, 2017 [12] |
Lancet Infect Dis |
· NNV = about 20 (not specified) Severe safety signals (no significant differences): (influenza vaccinated vs. placebo) · Maternal deaths: 3 vs. 5 in the abstract (2 vs 5 in STable2) · Miscarriages: 5 vs. 3 · Stillbirths: 33 vs. 31 · Infant deaths: 61 vs. 50 in the abstract (60 vs 51 in STable2) · Congenital defects: 20 vs. 18 |
|
Zaman |
Mother’sGift project, 2008 [19] |
New Engl J Med |
· NNV = 17 Severe safety signals (no significant differences): (influenza vaccinated vs. pneumococcal) · Maternal deaths: 0 vs. 0 · Fetal + infant deaths: 4 vs. 2 |
|
Tapia |
Mali trial, 2016 [21] |
Lancet Infect Dis |
· NNV = 99 Severe safety signals (no significant differences, except presumed serious neonatal infections and total SAEs): (influenza vaccinated vs. meningococcal) · Maternal deaths: 1 vs. 0 (in Table S6: SAEs in participating women ‘at any time after vaccination until 6 months post-partum’), but 2 vs. 3 (in the article) · Stillbirths: 24 vs. 30 · Infant deaths: 52 vs. 37 · Presumed serious neonatal infections: 60 vs. 37 (p=0.02) · Total SAEs: 225 vs. 175 (p=0.01); excluding major congenital malformations 219 vs. 171 (p=0.015) |
|
Simões |
Matflu (reanalysis) 2019 [22] |
Vaccine |
Serious safety signals (no significant differences): (influenza vaccinated vs. placebo) · Fetal death: 16 vs. 13 (RR 1.21) · Preterm birth: 100 vs. 80 (RR 1.22) · Low birth weight: 123 vs. 107 (RR 1.21) · Small for gestational age: 156 vs. 138 (RR 1.21) |
Round 2
Reviewer 3 Report
The revision is a vast improvement over the original submission.